# Person-Centered Health Promotion: Learning from 10 Years of Practice within Long Term Conditions

**DOI:** 10.3390/healthcare9040439

**Published:** 2021-04-08

**Authors:** John Downey, Saul Bloxham, Ben Jane, Joseph D. Layden, Sam Vaughan

**Affiliations:** School of Sport, Health and Wellbeing, Plymouth Marjon University, Plymouth PL6 8BH, UK; sbloxham@marjon.ac.uk (S.B.); bjane@marjon.ac.uk (B.J.); Jlayden@marjon.ac.uk (J.D.L.); Svaughan@marjon.ac.uk (S.V.)

**Keywords:** person-centered care, health promotion, implementation, behavior change, primary care

## Abstract

The utilization of person-centered care is highlighted as essential for health promotion, yet implementation has been inconsistent and multiple issues remain. There is a dearth of applied research exploring the facets of successful implementation. In this paper, a person-centered wellbeing program spanning various groups is discussed, outlining the central principles that have allowed for successful outcomes. Ten years of pragmatic pre–post service evaluation have shown consistent improvement in measures of functional capacity and wellbeing. The method for this paper is a reflective exploration of the theory and practices that can explain the continual improvement the clinics have achieved over 10 years. Core principles relate to connecting with people, connecting through groups, and connecting with self. The operationalization and theoretical explanation of these principles is outlined. The discussion of these principles posits essential factors to prioritize to advance the implementation of person-centered care in health promotion for long-term conditions.

## 1. Introduction

Non-communicable diseases (NCDs) are diseases that are not acquired through transmission. NCDs represent the largest threat to global mortality and an unsustainable demand on health services worldwide [1]. The dominant NCDs are chronic, develop over time, require self-management and include cancer, diabetes, cardiovascular disease, and respiratory disease [2]. There is substantial overlap in the label “NCDs” and “long-term conditions”, as both define disorders which are ongoing, worsen over time, and are typically mediated through poor lifestyle choices. The label “long-term conditions”, however, also captures interrelated disorders that are not defined as NCDs, for example, chronic pain. Moreover, and in line with our approach below, the term “long-term conditions” provides a more appropriate lexicon for these populations. Long-term conditions and unhealthy lifestyle behaviors tend to cluster in low socioeconomic groups. The term NCDs conflates the role of socioeconomic determinants of health. The risk-taking behaviors associated with long-term conditions are arguably communicable as they are passed across generations, which challenges the term NCDs [3].

Long-term conditions are a pressing challenge for contemporary healthcare. Long-term conditions are responsible for half of the global deaths in those over 40 years of age [4]. By the age of 50 years, half the United Kingdom will have one long-term condition. Worryingly, long-term conditions demonstrate a progressive trend and there is now a high prevalence of people with three or more long-term conditions. This multimorbid status leads to a decreased quality of life, increased risk of premature death and an unsustainable demand on health and social care systems [5]. Long-term conditions also lead to an expanding proportion of people who are less functionally capable, have lower health literacy, and respond poorly to usual care [6,7].

Yet, much of the burden of long-term conditions could be prevented through engagement in healthy lifestyle choices [8]. Patient- or person-centered care (PCC) is a fundamental practice to activate people in self-care and develop self-management skills, especially in lifestyle management [9]. Patient activation involves people valuing behavior change and acquiring the knowledge, skills, and confidence to adopt healthy lifestyle behaviors. Achieving patient activation leads to better health outcomes, improved care experiences, and a decreased health resource wastage [10]. PCC is broadly defined as a philosophy that encourages a shared control of care and a focus on the person as a whole. Fundamentally, the approach rejects an illness-centered approach and values the person’s experiences, beliefs, personhood, and identity [11]. In PCC, the person’s values and experiences are utilized, with the clinician’s enterprise, to develop a collaborative care plan [6]. Despite the enthusiasm associated with PCC across academia, healthcare, and policy, the evidence for positive changes to patient satisfaction and health outcomes are mixed [12]. The continued advocation for healthcare to be person-centered, and attempts to operationalize it since the 1950s, have not led to the widespread translation of PCC to practice. This organizational change is immense and multiple barriers exist [13,14]. 

Achieving the implementation of PCC remains elusive and despite the development of numerous conceptual frameworks, essential competencies, and ongoing training, PCC in usual care remains rare [15]. A partial explanation of the advocation for patient-centeredness but limited organizational change is the challenges of real-life care. Clinical care relies on human interpretation, social meaning, volition, interpersonal characteristics, and organizational context [16]. For the most part, PCC lacks an awareness of self and relies heavily on unconscious processes [17]. This contrasts with many conceptual frameworks which strip out context and reduce practice to a set of guidelines [18,19]. Organizational change requires clear, relevant, and pragmatic information. The inability of written guidelines to capture the complexities of real life means PCC ends up as a tick box with a paternalistic prescription model that conflates health promotion and is inappropriate to support the adoption of PCC [20]. This array of contextual factors means organizations are unaware of the essential conditions and practices which will achieve PCC. Despite renewed policy commitment to utilize PCC, and social and behavioral science [21,22,23] the translation to practice remains poor. There have been recent calls to provide examples from applied practice outlining the successful factors of PCC adoption [14].

If there is to be widespread change, organizations and clinicians require practical information on what facets are paramount to achieve PCC [15]. Therefore, reflections from successful practice presents useful learning and generalizability [24]. Reflections from successful practice can evaluate and integrate tacit knowledge from a range of stakeholders and unearth prudent information that is cognizant of the complexities of real-life care. This paper utilizes knowledge gained from experience to build on the current understanding of the necessary elements to achieve PCC in practice. This paper will outline concepts that we discussed as a team and then immersed in the theoretical literature to re-describe our experiences using evidence-based principles and tools. The Marjon Health and Wellbeing (MHW) approach is a person-centered wellbeing intervention which is utilized across numerous long-term conditions. The aim of this paper is to present the pragmatic learning from 10 years of practice, the agreed clinic principles and assigned theoretical underpinnings of practice, and the operational tools that have been used routinely through the years.

## 2. Materials and Methods

### 2.1. Clinic Delivery

The health promotion clinics started in 2009, and all services now adopt the same overarching principles and structure yet retain a degree of flexibility to tailor delivery to individual groups and commissioners. Typically, groups meet for two hours per week for a period of 4–6 weeks and engage with a multidisciplinary approach that encourages self-management of health issues. During each program, participants are introduced to physical activity, mindfulness, cognitive behavioral assumptions, sleep hygiene, and healthy eating with appropriate signposting (see [25] for intervention overview and data procedures and processing). The evaluation of programs is a pragmatic pre–post design. Previous research has appraised discrete cohorts from an empirical stance [25,26]. The current paper proposes a model which explains the continual, successful pattern of outcomes. The quantitative findings are provided for visual support alone and do not claim rigor, as defined in the experimental research design language, as they originate from a service evaluation.

### 2.2. Reflection as Data

The aspects presented below were developed from over 10 years of practice, numerous research papers, multiple iterations of commissioned services, and ongoing service evaluations across multiple long-term conditions. The approach aligns with the idea that practice is a form of research and reflection generates new knowledge that learns from the complex nature of real world environments [27]. The knowledge presented is from an accumulation of learning in the form of the recurring conditions needed for PCC collected phenomenologically rather than empirically [28]. All the authors’ experiences of interacting with patients, developing services, speaking with students who supported the clinics, and feedback from external partners shaped a consensus which we present in the below section. Over the last year, a more formal reflective dialogue with program architects, service users, and clinic leads was undertaken to consolidate the essential practices and principles that lead to successful outcomes and PCC translation. The principles outlined below are supported by service user extracts from the service evaluation documents.

### 2.3. Utilizing Theory to Re-Describe Learning

Facets that were consistently experienced, discussed, and highlighted by other stakeholders were categorized using inductive labels. Within these broad areas, iterative conversations provided the essence of practice. The theoretical underpinning of PCC is rarely articulated, principally as PCC relates to more than just the sharing of power between a clinician and a service user. Therefore, there is no single unifying mechanism on how PCC may be translated or indeed work in practice. We took the broad areas highlighted from our experiences and consulted the literature to re-describe core principles in a theoretical sense, and articulate routine practices that have endured across the 10 years in evidence-based terms. We acknowledge that there may be other ways to abstract our learning but chose aspects of theory that align to the philosophy of PCC [14,29] and resonate with our interpretation of the core conditions needed for PCC.

## 3. Results and Discussion

In line with the call to present examples of successful applied PCC practice, the current paper outlines the core facets which have allowed for the implementation of PCC. It is argued that principles covering connecting with people, connecting through groups, and connecting with self, encapsulate the success of the programs. Practices typically focus on increasing autonomous motivation, humanism, enhancing perceptions of control, providing and facilitating social support and group identity changes, and transformative learning. Many of these practices are known to enhance adherence to services in long-term conditions [30].

### 3.1. Dataset

Since its initiation in 2009, 1230 people have attended the cancer and chronic pain programs. Although not the main purpose of this paper, the initial clinics did have internal university ethics for evaluation purposes and subsequently all individuals gave written consent for their data to be gathered for evaluation. Due to the history of the programs (pilots, changes in funders, dynamic resource allocation) and evolving metrics over time, the completeness of the quantitative data has been impacted and experimental designs have not been possible. Despite varying sample sizes across the variables, the service evaluation showed significant improvements in all performance measures, including aerobic fitness (7%), handgrip (3%), and total weekly energy expenditure (59%) identified through a range of dependent *t*-tests on IBM SPSS Statistics 26. Disease-specific outcomes also indicated positive changes in functional capacity. Quality of life significantly improved for cancer patients. Perceived disability due to back pain showed a 16% reduction, and back flexion and extension endurance increased by approximately 21% and 32%, respectively. Table 1 outlines the characteristics of the clinic attendees including the Index of Multiple Deprivation, which is an English measure of socio-economic status using an individual’s home address. Thise data is for illustration purposes alone and provides validation of the principles and practices discussed, as it indicates that the service delivery is achieving successful outcomes.

### 3.2. Connecting with People

#### 3.2.1. Underpinning Theory

The overarching culture is one of humanistic practice and operates to ensure the service user’s values and preferences guide care via an unconditional positive regard for them [31]. The unconditional positive regard influences change through a natural actualizing theory in humans [32]. However, the MHW acknowledges that compassionate care is only one aspect of PCC. The organizational culture provides the epicenter for the approach as it does not have an enduring obligation to medical values and there is an intentional commitment to uphold the focus on the person holistically across multiple staff roles [33]. The adoption of practice based on mutual respect and person-led discussions provides an increased contextualization of care and acceptability for individuals increasing self-determined motivation. The setting mitigates many of the medical assumptions through a demedicalization of practice. The work operates to extinguish expert–patient assumptions about knowledge and encourages people to explore non-prescribed treatment options within a setting that decreases medical social control [34]. This setting allows for genuine humanistic care, which is acknowledged by people, as demonstrated in the below quote.

“I have taken back up badminton and cricket for the first time in 10 years. The staff work specific to the individual and they have the human touch” [which is] “motivating and confidence boasting” (cancer service user).

#### 3.2.2. Practices

The spirit of motivational interviewing (MI) provides a tangible way to operationalize humanistic care [35]. Collaboration, compassion, and acceptance are implemented to increase self-actualization and to shed a dysfunction-centric approach. Introducing patients to a range of opportunities and tools, along with an ongoing information exchange and thorough group reflection, is the primary way evocation is operated. Practices to support intrinsic motivation include the provision of choice, providing rationale for advice and activities, exposing patients to new challenges and environments and setting homework, providing positive feedback, and developing social relationships and a group identity [36]. Groups are co-delivered by university students which is envisaged to demedicalize the programs by creating equal partnerships and minimizing power relationships [37]. Support workers provide individuals with a contact point and safety net to try things, feel genuine devotion of care, understand the new environment, converse about their life and barriers, and gain basic skills in new environments [38]. Lastly, the setting provides an antidote to the medical model as it has a history of collaborative teaching and a culture that supports individual prosperity. These provisions address key mechanisms of successful care including patients feeling believed, supported, encouraged, and in control during their educational experience [39]. The following quote illustrates this aspect of practice. 

“The instructors led to the success of the program as they were pleasant and approachable, and we worked as a team. They were supportive and at no time judgmental which often deters the seriously overweight person” (back pain service user).

### 3.3. Connecting through Groups

#### 3.3.1. Underpinning Theory

The small group format draws on the principles of social identity theory, which proposes that increasing social connectiveness shapes attitudes, cognitions, and behaviors and valued membership can increase wellbeing [40]. Group structural elements impact the internalization of social identity, and the creation of opportunities for reappraisal and interaction can modify the perceived norms for certain identities within specific contexts [41]. Additionally, the programs operate education through experiences, modelling, and group learning to target attitudes, self-efficacy, and norms as per the social cognitive theory [42]. The following quote articulates how individuals were benefiting from the group environment.

“The group aspects with people undergoing the same difficulties has given me great confidence to venture into the gym and take each day as it comes and not feel like a failure” (cancer service user).

#### 3.3.2. Practices

Connections and group belonging need facilitating and key features are implemented in the MHW. Groups sizes tend to be 4–10, occur in same condition cohorts, have planned and frequent breaks for interaction and sharing, and have personable and credible facilitators. Practices to activate social theoretical constructs crosses over with previous practices such as MI and autonomous care. In brief, the facilitator adopts techniques including checking understanding, encouraging contributions, outlining sessions, paraphrasing, providing examples, discussing outcome expectations, promoting motivation and confidence, prompting social learning, validation and social comparisons [43]. These practices operationalize many well-known “behavior change techniques” that are also imbedded in the programs. For example, participants are taken into a gym environment and offered a choice of what they would like to experiment with. Students demonstrate activities, persuade individuals about capability, and provide a reflective opportunity for people to experience exercise in a safe environment supporting biofeedback, reattribution of the cause of pain and discomfort, and information on the consequences of behaviors. The staff’s role in facilitating the group was a key aspect of the work as recognized in the quote below.

“Staff spent time and put effort into working as a group but also on an individual level. Realizing you are not alone by working in a group was helpful” (back pain service user).

### 3.4. Connecting with Self

#### 3.4.1. Theoretical Underpinning

The pedagogical approach mirrors transformational learning to build autonomous and liberated individuals. Transformative learning leads to a change in an individual’s embodied frames of reference [44]. The format provides open experiences for individuals to challenge their identity; as individuals tend to reject ideas that do not correspond with their preconceptions [45,46]. In line with a previous facet of theory [34] there is not a paternalistic drive to make people conform to medical assumptions, instead the approach looks to enhance human flourishing and perceived control of their own wellbeing. There is a focus on holism and activating the patient in their health via shifts in self-concept [47]. The outcome of the approach is to increase the learner’s capacity to critically evaluate experiences and take action [46].

#### 3.4.2. Practices

The MHW provides an avenue for individuals to reflect and challenge thoughts, acknowledge automatic thought processes, develop new ideas and experiment in a safe environment [48]. Additionally, the programs generate an environment where individuals are encouraged to be active members in the education provisions. The lead practitioner pledges an explicit commitment to practice in ways that support the patient’s identity by addressing the needs for attachment, comfort, occupation, and inclusion, increasing the patient’s self-worth and a sense of feeling valued [11]. The programs involve multicomponent experiential learning, where individuals are supported to experiment with a range of tools that they are encouraged to try and either adopt or reject based on their preference. The accumulation of knowledge, skills, and confidence is achieved through a broad lens of elements that may be useful for their health.

“I feel really well, and I think that’s due to this course and helping me understand a lot more about pain, about relaxation, about diet as well, and I feel confident about exercising” whereas “Mostly before I have been given painkillers and sent to the physiotherapists who say they can’t do much, just keep taking the painkillers. Now [this program] has opened up a whole new avenue of how to deal with pain, using different gym equipment… boxing… things I would never have dreamed of doing before. It has opened up a whole new aspect of dealing with it and coping with it [back pain]” (back pain service user).

## 4. Conclusions

Despite the enthusiasm to transition healthcare to a person-centered model, routine practice has not modified on a large scale. The discussion of the pragmatic, but essential, facets to operate PCC is underexplored. Due to the diversity of how PCC is interpreted, and the range of domains it possesses, its implementation is challenging. Reflections from successful practice are prudent as they acknowledge the complexity of real-world PCC utilization. What is presented here are the enduring facets of PCC which are a priority to achieve the envisaged outcomes of PCC within health promotion. The anticipated outcomes from PCC include patient activation, improved patient satisfaction, lifestyle behavior change, and improved health. This paper provides an outline on what should be prioritized for PCC implementation to support health promotion. The model outlines central factors that can mitigate the challenges in PCC. Firstly, an organizational commitment to, and culture conducive of, PCC is needed, which can enhance the operation of humanistic practices and a demedicalization of care. Staff should plan to enhance self-determined motivation in participants. Socialization and small group work support changes in wellbeing and identity and help deconstruct typical medical/patient power relationships. This paper presents key design and facilitator approaches that have endured through the years, which resonates with the literature [43,49]. Lastly, services should include transformative learning practices, creating an established pedagogical template to empower service user’s to rehearse elements, along with reflecting on and re-evaluating aspects that may be important to their health. The model provides pragmatic modifications that are needed to initiate an advantageous shift in PCC and behavioral science implementation in the treatment of long-term conditions.

If PCC is to supersede the biomedical model, there needs to be clear examples of success and a pragmatic roadmap of how success can be achieved. PCC is a diverse term which is often misinterpreted. This presentation of the practice and theory underpinning our model that works through connecting with people, connecting through groups, and connecting with self is a tangible and flexible system to facilitate organizations to challenge their dominant care paradigm. The model provides a system to enhance the fidelity to PCC which has shown, in our case, to reap the rewards that have been promised since the 1950s.

## Figures and Tables

**Table 1 healthcare-09-00439-t001:** Demographics of the program participants.

Variable	Outcome
Total people	1230
Age	52 ± 14.6
Female	61.70%
Index of Multiple Deprivation	12,960 ± 8988

## Data Availability

Original data from the empirical work are available on request. The current paper was a narrative concept paper and only outlined data for visual use.

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
