# Peer review of "Person-Centered Health Promotion: Learning from 10 Years of Practice within Long Term Conditions"

_healthcare, 2021, doi:10.3390/healthcare9040439_

Round 1

Reviewer 1 Report

As a standard format article, the current paper lacks a clear research question, and therefore I have reservations in terms of how it could advance scientific knowledge. At the moment, it feels more like a magazine article than a scientific paper. Perhaps it would serve better as a "commentary" or similar format?

Author Response

That you for the prompt response. We acknowledge that the format looks similar to a commentary in places, and we would be open to reshaping it to be more like a commentary if the editor thinks this would be a useful contribution for the special edition in health and exercise promotion. Although the results/discussion seem like a commentary, the description of key principles and procedures using a theoretical lens is based off the methodological stance that practice and reflection are a form of research. The exploration of theory was not based on a commentary of the literature but our experiences and verification with a range of stakeholders .

Reviewer 2 Report

I am confused.  The authors are saying this is an essay?  I do not see any indication of that in the first round of reviews.  Can you clarify?

Author Response

Apologies for the oversight, essay was not the best word to use in the
text of our response to you. Please read it as "paper".

If however, you are also raising this issue because of confusion around
the methodological approach in the initial submission then, as outlined
in the first submission, and expanded in the second submission, this is
not a study with an experimental design. Our approach does, however,
make an original contribution by using our experiences, reflections from
practice, and discussions across various stakeholders to propose a
theoretical model of successful PCC in this domain.

This manuscript is a resubmission of an earlier submission. The following is a list of the peer review reports and author responses from that submission.

Round 1

Reviewer 1 Report

The paper under review described authors' learnings from a programme that has been running for 10 years. The paper was largely written in clear language (except for 1 sentence which I had difficulty understanding - please see comment below), with concepts articulated clearly. Please find my comments to the paper below:

  1. I was not familiar with PCC prior to reading this manuscript. However, I'm afraid the concept is still very vague to me after reading the paper. Given this is the central concept of the paper, I feel the description of the key concept should probably be improved. A bit more history/examples of the approach would be welcomed.
  2. To me, the current version of the manuscript is lacking in a clear objective/hypothesis. I understand this paper was not written with a goal of testing any hypotheses, but it still feels too descriptive to me. 
  3. The paper also appears to be lacking in any sort of analytical input. Instead, I feel this was simply a list of facts and some views from the authors. 
  4. What are the strengths of the study and how does the paper advance scientific knowledge or theory? Currently, I feel arguments to these points are lacking.

Specific comments:

  1. What are "long term conditions" (first line of intro read "long termS condition", I believe that's a typo)? I think many people would know, but I feel it would be best to define it at the start of the paper.
  2. The authors listed a number of areas in which PCC is challenged in Paragraph 2. Which of those were valid and which of those could be refuted? Simply from reading this paragraph, I almost felt the authors were against this approach.
  3. The authors mentioned that "theoretical underpinning of PCC is rarely articulated" (line 51), and this paper aims to "assign theoretical underpinnings of practice" (line 75). In combination, does that mean the approach actually doesn't have any theoretical basis?
  4. What is "Index of Multiple Deprivation"? (Table 1)
  5. "Individuals aspects... survey components" (Line 113). I tried reading this line multiple times and I struggle to understand it.

Reviewer 2 Report

This is an important topic, and I appreciate the authors work in categorizing Person Centered care in this framework.  Often we will use the term chronic disease vs. long term.  I don't believe they really discussed how they approached the literature search to help them come up with their framework.  They also mention their experience using the model, but don't really show any results or patient feedback which would be interesting.

Reviewer 3 Report

The authors present a very interesting concept, but the organization of the manuscript makes it very difficult to review. 

Introduction - This was informative and appropriate.  However, I am not sure why the authors used the term 'long term conditions' rather than 'chronic conditions'.  Some sort of definition would be helpful.

Methods - This is a very short section and does not provide much information regarding the study design, details of group meetings, data collected, how participants were recruited etc etc. Some the study giving ethical approval? Did the participants give informed consent or was this a retrospective analysis with de-identified data?  Were any statistical analyses performed?

Results and Discussion - Much of this section seems to be methods?  Where is the data/results?  There are no figures and only one table of demographics?  What did the study find and how dies the discussion related to them?

At best, the manuscript needs a complete re-organisation so the reader can understand what was done and what was found.